# Therapeutic Potential of Human Stem Cell Implantation in Alzheimer’s Disease

**DOI:** 10.3390/ijms221810151

**Published:** 2021-09-21

**Authors:** Hau Jun Chan, Jaydeep Roy, George Lim Tipoe, Man-Lung Fung, Lee Wei Lim

**Affiliations:** School of Biomedical, Sciences, Li Ka Shing Faculty of Medicine, The University of Hong Kong, Hong Kong, China; lvnigel@connect.hku.hk (H.J.C.); yanshree@connect.hku.hk (Y.); jaydeep@connect.hku.hk (J.R.); tgeorge@hku.hk (G.L.T.); fungml@hku.hk (M.-L.F.)

**Keywords:** stem cells, neurogenesis, Alzheimer’s disease, stem cell therapy, neural stem cells, neurodegenerative disease, embryonic stem cells, mesenchymal stem cells, stem cell transplantation, Induced pluripotent stem cells

## Abstract

Alzheimer’s disease (AD) is a progressive debilitating neurodegenerative disease and the most common form of dementia in the older population. At present, there is no definitive effective treatment for AD. Therefore, researchers are now looking at stem cell therapy as a possible treatment for AD, but whether stem cells are safe and effective in humans is still not clear. In this narrative review, we discuss both preclinical studies and clinical trials on the therapeutic potential of human stem cells in AD. Preclinical studies have successfully differentiated stem cells into neurons in vitro, indicating the potential viability of stem cell therapy in neurodegenerative diseases. Preclinical studies have also shown that stem cell therapy is safe and effective in improving cognitive performance in animal models, as demonstrated in the Morris water maze test and novel object recognition test. Although few clinical trials have been completed and many trials are still in phase I and II, the initial results confirm the outcomes of the preclinical studies. However, limitations like rejection, tumorigenicity, and ethical issues are still barriers to the advancement of stem cell therapy. In conclusion, the use of stem cells in the treatment of AD shows promise in terms of effectiveness and safety.

## 1. Introduction

Alzheimer’s Disease (AD) is a common neurodegenerative disease, accounting for 60–70% of neurocognitive disorder-related illnesses [1]. Neurocognitive disorder is an umbrella term covering a wide range of cognitive disorders that are increasingly common in the aging population, leading to considerable economic and societal burdens [2]. In 2015, 5.1 million individuals over the age of 65 were diagnosed with clinical AD in the United States, and over 47 million people worldwide were estimated to have neurocognitive disorders [3]. The number of cases is predicted to increase to 13.8 million in the United States and to more than 130 million worldwide by 2050 [4,5]. Key characteristic symptoms of AD include various cognitive impairments such as difficulty in remembering or recalling recent events [6]. The symptoms of AD can be categorized as mild, moderate, or severe. Individuals with mild AD symptoms are more likely to get lost, have poor judgment leading to bad decisions, increased anxiety, and personality changes. Individuals with moderate AD symptoms lose the ability to learn new things, have language problems such as reading and organizing thoughts, and have difficulty in recognizing family members. Individuals with severe AD symptoms experience weight loss, skin infections, difficulty in swallowing, and lose the ability to communicate [1].

One of the major risk factors for AD is age. A study by Guerreiro et al. identified a locus on chromosome 17 associated with onset age, with a specific variant CCL11 suspected to be associated with the onset of AD [7]. Recent studies have established an association between depression and AD [8]. In 2021, Tanaka et al. demonstrated a link between AD and late-life depression by using resting-state functional magnetic resonance imaging. The dissociated functional connectivity pattern with decreased posterior default mode network (DMN) and increased anterior DMN is commonly observed in AD and late-life depression [9]. Aside from depression, gender is also a factor for AD risk, with a greater prevalence in females than males [10,11,12,13,14]. Some studies, such as Scheyer et al., have suggested a possible link between menopause and AD [15,16,17]. More recently, Sini et al. indicated several environmental factors could lead to AD. They found that cyanobacteria, present in natural water samples, can produce four classes of neurotoxins: saxitoxins, ciguatoxins, anatoxins, and β-N-methylamino-L-alanine (L-BMAA), which can all lead to an increased risk of AD [18]. Adding to the neurological disorders elicited by AD, some infections are more prone to appear in AD patients. Infections associated with AD include pneumonia, oral herpes, and spirochete bacterial infections causing Lyme disease and gum disease. These associated infectious illnesses can lead to chronic inflammation and eventually death [19]. Nonetheless, the life expectancy of AD patients following diagnosis can be up to 9 years [6].

The high rate of neuronal loss in AD patients can make treating this disorder difficult [20]. Conventional drug treatments that can restore brain tissue and improve cognitive functions can have undesirable side effects [21]. Stem cell therapy is a potential AD therapy that can overcome these undesirable outcomes. Stem cell therapy as a treatment for neurological disorders has been gaining interest in the field. With a self-renewal property, stem cells can go through numerous cycles of division and growth [22]. Stem cells can also differentiate into various specialized cell types [23]. These properties make stem cells a possible treatment option for AD by aiding in the proliferation and repair of damaged brain tissues [21]. Recent preclinical studies on stem cell therapy for AD have been promising and clinical trials on humans are ongoing. In this review, we first provide a brief description of AD etiology. Next, we summarize the current as well as the alternative treatments for AD. We further discuss the mechanisms of stem cells and their applications in preclinical studies and clinical trials. Lastly, we explore the limitations and possible future applications of stem cell therapy in AD.

## 2. Etiology of Alzheimer’s Disease

Alzheimer’s disease is a neurodegenerative disorder involving the accumulation of senile plaques and neurofibrillary tangles. Various neuropsychological, clinical, neuroimaging, and laboratory techniques are currently used to diagnose AD [24]. Individuals with AD exhibit key neuropathological changes including the deposition of extracellular beta-amyloid peptides as senile plaques [25,26,27,28] and accumulation of intracellular tau-containing neurofibrillary tangles in the brain [29,30]. All forms of AD have been found to involve senile plaques and almost all of them share increased production and decreased clearance of amyloid-beta peptides. However, certain mutations such as the “Arctic” mutation and “Osaka” mutation not only show slightly increased levels of amyloid-beta peptides, but also increased protofibrils and aggregations, respectively [31,32]. Amyloid-beta peptides are released by the cleavage of amyloid precursor protein (APP) [6] by two enzymes, beta-secretase and gamma-secretase [33]. Gamma-secretase is closely associated with presenilin (PSEN), and mutations in PSEN1 and PSEN2 can increase the synthesis of amyloid-beta [34,35]. However, the exact mechanism of how amyloid-beta proteins cause AD is not fully understood. Scientists have proposed that the aggregation and deposition amyloid-beta plaques in the brain activate neurodegeneration, leading to key symptoms such as memory loss [36,37]. Another prominent protein in the pathogenesis of AD is tau [38], which is a microtubule-associated protein that aids in the assembly and stabilization of microtubules. In AD, tau becomes hyperphosphorylated and assembles into paired helical filaments that detach from microtubules and attach to other tau molecules. They form threads that aggregate into neurofibrillary tangles, which block neuronal transport, preventing the movement of molecules and nutrients from the cell body to dendrites and the axon, leading to disrupted synaptic communication [39].

Besides the main theories, there are several other hypotheses of the pathogenesis of AD. A current theory proposes a three-part mechanism involving decreased levels of blood lactic acid, folic acid, and increased levels of blood ceramide and adipokines [40]. These three mechanisms result in age-related characteristics, such as decreased muscle mass, change in diet, and increased visceral fat, respectively.

Lactic acid is synthesized in muscle cells and blood cells due to low oxygen levels. It is vital in supplying energy to brain cells like astrocytes and pericytes [41]. Inadequate levels of lactic acid can result in damage to the endothelial cells and pericytes in the blood−brain barrier, leading to brain damage. Studies have also shown that dietary changes in the aging population may contribute to inadequate folate intake [42,43,44,45]. As folate helps maintain the blood−brain barrier in protecting endothelial cells [46], a diet that contains adequate folate could aid in delaying the onset of AD and slow the cognitive decline in older adults [46,47]. High levels of ceramide have been discovered in the brain and blood of AD patients [48], which indicates that elevated levels of ceramide could lead to a greater risk of AD [49,50,51,52]. Ceramide induces oxidative stress and increases NADPH oxidase activity outside the plasma membrane of macrophages in the brain. This can increase hydrogen peroxide production, leading to damaged neurons in the brain [52]. Inflammatory adipokines can be secreted into the blood by visceral fat, causing arthritis, type 2 diabetes, heart disease, and neurological problems [50]. Visfatin, a type of adipokine, interacts with xanthine oxidase and NADH oxidase to boost the production of oxygen radicals in the capillary lumen. This can lead to oxidative damage in the blood−brain barrier, eventually damaging neurons [40]. In 2020, Tanaka et al. discovered increased levels of pro-inflammatory cytokines and neurotoxic kynurenines in neurodegenerative diseases including AD, which can damage the neuronal structure in the brain [53].

## 3. Current Treatments for Alzheimer’s Disease

Although some treatments for AD are available, their effectiveness is questionable due to the nature of AD [54]. Alzheimer’s disease is a multifactorial disease and is diagnosed through its clinical manifestation and underlying brain pathology. Typically, a disease should contain the following three basic factors: 1. An established biological cause, 2. A specific set of symptoms, and 3. Consistent anatomy changes. However, AD does not have an established cause and the symptoms are not well-defined. Moreover, the exact biological cause is not known, and the differential symptoms can vary from person to person, making it challenging to find a cure for AD. A study conducted by Salomone et al. noted challenges in treating AD due to the ineffectiveness of drug therapies [55]. Another review by Rijpma et al. also concluded that no single drug and nutrient-based therapy was clinically effective against AD [56]. Nevertheless, there are currently several drugs under clinical trial, and some have even been approved as treatments for AD. Possible interventions targeting metabolites and enzymes in the kynurenine pathway of tryptophan metabolism are also under investigation [57].

The currently approved drugs mainly alleviate symptoms and slow down the disease progression [57,58,59,60,61]. The N-methyl-D-aspartic acid (NMDA) receptor inhibitor and acetylcholinesterase inhibitor (AChEI) are two classes of approved medications for clinical use [62]. Acetylcholinesterase inhibitors (AChEls) act by inhibiting synaptic cleft cholinesterase from breaking down acetylcholine, thereby increasing cholinergic transmission in the cerebral cortex and basal forebrain [63,64]. Donepezil, rivastigmine, and galantamine are some examples of AChEI [65]. Another drug class effective against AD is NMDA receptor inhibitors. These drugs reduce the excitotoxicity generated by NMDA receptors excitation and protect the neuronal cells in the brain [66]. Memantines, a commonly used NMDA receptor antagonist, is also useful in relieving some symptoms [67]. Drugs with mechanisms of action relating to NMDA receptors are also under clinical trials [68,69]. AVP786 is a weak NMDA receptor antagonist and is currently under phase 3 trials; however, many trials have revealed it is ineffective for treating AD [70]. Another drug, BI425809, is a co-agonist of NMDA receptors and is currently being tested in phase 2 clinical trials [71]. Besides the above two drugs, gavestinel and AXS-05 are also undergoing clinical trials [72,73]. In recent years, novel drug targets have also been established. Drugs such as Solanezumab, Aducanumab, and Crenezumab are monoclonal antibodies targeting Aβ peptide [74]. These antibodies can bind to Aβ peptides and help to clear excess amyloid plaque and reduce sunaptotoxicity, eventually leading to improved cognition in AD patients [75,76,77]. Other drugs like Anavex 2–73 and GV-971 work by blocking tau hyperphosphorylation to reduce AD pathology [78].

Another way to develop drugs against AD is to repurpose existing drugs for dementia. Moreover, other drugs can be repurposed, including diabetes agents and vitamins, as well as drugs for a wide range of diseases from cardiovascular to psychiatric disorders [79]. For example, there is evidence that antioxidants (e.g., vitamin E at a dose of 2000 IU/day) can delay functional impairments [67,80]. Masitinib, a tyrosine kinase inhibitor which was originally used as a treatment for mast cell tumors, is also suggested to have anti-dementia effects [81]. Zolpidem, a sedative−hypnotic medicine prescribed for insomnia, is also a promising drug for treating AD [57,82]. Besides drug treatments, exercise programs have been shown to help AD patients physically, but they do not improve cognitive functioning [83].

Scientists are also looking at novel targets and approaches against AD. Current research includes genetic instability, post-translational modification, and lipid metabolism related to long interspersed nuclear element-1, micro RNAs, and apolipoprotein E4, respectively [84,85,86,87,88,89,90,91]. Calmodulin-binding proteins associated with calcium homeostasis [92] have also been shown to have therapeutic potential against AD. Additionally, kynurenine analogues, which are NMDA receptor antagonists and antioxidants, can reduce neurotoxicity in AD patients [53,93,94,95,96,97,98]. In 2021, Ibos et al. suggested the presence of a sex-dependent hemodynamic compensatory mechanism could also be a potential therapeutic direction in AD [99]. Diet-wise, the use of nutraceutical compounds can also possibly play a prophylactic role in AD. Supplemental use of nutraceutical inositol was suggested to prevent the onset and progression of the cognitive impairment in AD [100].

## 4. Alternative Strategies for the Treatment of Alzheimer’s Disease

Currently, there is no effective treatment that can cure AD. Recent clinical studies suggest that electrical stimulation might improve memory functions when specific brain regions are stimulated. Of particular interest is a single-case report in which electrical stimulation was used to treat a patient with morbid obesity, in which the electrical brain stimulation unexpectedly evoked autobiographical memory episodes in the patient [101]. In animal studies, experimental data showed that memory functions could be enhanced by stimulating the medial prefrontal cortex [102,103,104,105], entorhinal cortex, and perifornical region [106,107,108]. It has also been shown to induce antidepressant-like effects in animal studies [109,110]. Nevertheless, without in-depth mechanisms of preclinical studies, it is still a very premature phase to draw any conclusion on whether electrical stimulation will be suitable as a treatment for patients with dementia.

Drug therapies for AD, mainly given on an individual basis, can only temporarily improve some symptoms, but cannot stop or slow down the neurodegenerative process [111]. The low efficacy of these drugs is exemplified in the high risk/benefit ratio of AChEls, where symptoms are only slightly improved when compared with a placebo [112]. Due to the low efficacy of current treatments, pharmaceutical companies and medical institutes have been actively seeking alternative therapies for AD, including stem cells transplantation.

## 5. Outline of the Review

The online PubMed database was searched for relevant articles between the years 2000 to 2021 in English using a Boolean operation with keywords “Alzheimer’s” AND “stem cell”. Relevant articles cited in the reference lists of the identified review articles were also included. The search found 153 preclinical studies with the majority of them published in 2020, and 708 clinical studies, of which 114 were also published in 2020. We used PubMed as it contains an extensive collection of indexed peer-reviewed journals. Another online database, ClinicalTrials.gov (accessed on 10 August 2010), was searched for relevant clinical trials using keywords “Alzheimer’s disease” and “stem cells”. We review selected preclinical studies and clinical trials of stem cell therapies for AD and discuss their efficacy and safety in AD patients. The mechanisms, therapeutic potential, and limitations of the stem cell therapies are also fully discussed.

## 6. Therapeutic Potential of Stem Cells

Rosenberg (1988) was the first to successful graft genetically modified cells in a damaged brain to protect cholinergic cells from dying [113]. However, this approach has drawbacks because it can only generate a small amount of tissue [114]. This pioneering study paved the way for stem cell implantation as a therapy for neurocognitive disorders. Another study in 1991 showed the engraftment of neuronal cells in specific brain regions was able to provide temporary relief in Parkinson’s disease by increasing the concentration of neurotransmitters, but it was unable to treat the disease [115].

A preclinical study by Wang et al. in 2006 investigated the effects of NSC transplantation in the cortex of an AD mouse model [116]. They transplanted mouse embryonic stem cell-derived neurospheres into the barrel field of the S1 cortex and frontal association cortex of C57BL/6 mice that had lesions of the nucleus basalis of Meynert (NBM) induced by ibotenic acid. Another group of mice receiving only embryonic stem cells served as the control. After 12 weeks of transplantation, mice were subjected to behavioral testing, which demonstrated the transplanted neurospheres in the cortex had survived and were able to produce more ChAT and serotonin-positive neurons within and near the implanted grafts. They found the mice with implanted neurospheres had significantly decreased working memory errors in the eight-direction maze test. On the contrary, treatment with embryonic stem cells in the control group led to teratomas, and no neurons were expressed, resulting in rapidly deteriorating working memory. In another study, neurospheres cultured from free floating clusters of NSCs were found to produce a positive effect in ameliorating the symptoms of AD in mice [117]. The form taken by the neural stem cells was found to greatly impact their effectiveness, as implanted embryonic stem cells not only had diminished positive effects, but also formed adverse teratomas. Various other similar investigations on animals were subsequently conducted to understand the effect of implanted stem cells in treating AD.

Later, scientists developed another form of stem cell therapy that involved pharmacological activation of endogenous neural stem cells and progenitors [118]. One example is epidermal growth factor (EGF) that is reported to be a critical mitogen for regulating neural stem cell growth and maintenance [119]. In stroke animals, intraventricularly administered EGF was able to increase neuronal differentiation rate in the striatum [120]. Teramoto et al. injected EGF and albumin intravenously in a mouse model of cerebral ischemia, which resulted in increased neuronal replacement by 100-fold [120]. They also found that newly developed immature neurons had migrated to the lesion site and differentiated into mature neurons, replacing more than 20% of the lost interneurons within 13 weeks. Although this method of stem cell therapy has not been widely applied in AD trials, the above experiment shows it has immense potential for treating AD.

## 7. Preclinical Research on Stem Cell Therapies in AD

Stem cell transplantation is a relatively new form of treatment in AD. To date, many studies have been conducted on stem cell therapies as a treatment for AD, but they have mostly been preclinical research. We selected several important preclinical studies to evaluate the effectiveness of stem cells therapies in AD rodent models (Table 1).

For in vitro experiments, stem cells or normal cells are usually cultured to test the direct effect of certain medications on cells and tissues [121]. For example, neutrospheres can be formed from stem cells and are used to study extrinsic stimuli present in the neuronal microenvironment [122,123]. For stem cell therapy, stem cells need to be directly transplanted in vivo to elicit the effects. Thus, in vitro experiments are very important to ensure the safety and primary effectiveness of stem cell therapies. Early studies of stem cell therapy for AD by Farshed et al. investigated the generation of neurons using mouse embryonic stem cells (ESCs) [124]. After deriving ESCs from the C57BL/6 mice, stem cells were grown on a feeder layer of primary mouse embryonic fibroblasts in a tissue culture flask containing the Dulbecco’s modified eagle medium. Ref. [125] The neuronal precursor cells (NPCs) were then successfully differentiated for transplantation. Liu et al. in 2013 also performed a similar experiment using human ESCs [126]. They maintained the stem cells similar to Baharvand et al. and differentiated them into neuroepithelial cells and eventually into cholinergic spinal motor neurons [127]. To improve the differentiation and incubation procedure, Cheng et al. suggested the incorporation of electrical stimulation to promote stem cell neural differentiation [128].

After several in vitro studies successfully differentiated stem cells into a neural lineage, research then focused on mice with simulated AD pathology. Most of the mice in these studies were injected with Aβ1—42 to elicit AD-related symptoms. Before investigating the effectiveness of stem cells on rodent models, the safety of stem cell therapy needs to be investigated [128]. In 2011, Ra et al., conducted both preclinical and clinical studies on the safety of stem cell therapy. The preclinical study adopted comprehensively designed experiments to investigate the safety of intravenous infusion of human adipose tissue-derived MSCs (hAdMSCs) in immunodeficient mice [110]. The results showed the highest cell dose of 2.5 × 10^8^ cells/kg body weight did not produce any adverse effects, and the SCID mice were healthy. A tumorigenicity test was performed in Balb/c-nu mice for 26 weeks, which showed the highest cell dose of 2 × 10^8^ cells/kg body weight did not cause any tumor development, which confirmed the safety of MSC-derived neural stem cells in these rodent models [110]. A study by Farshad et al. successfully grew and differentiated stem cells in their laboratory. The implanted stem cells did not lead to tumor formation in lesioned rats, indicating these stem cells were safe for transplantation [124]. Besides their safety, the effectiveness of stem cell therapy in lesioned rats has also been demonstrated. Liu et al. observed that stem cell transplantation in injured rats reduced latency to find a hidden platform in the Morris water maze (MWM) test. The transplanted rats also showed a shorter latency to avoid an aversion stimulus in the passive avoidance test. The accuracy of the above tests was confirmed in the open field test, which also showed little variation in the anxiety level that would otherwise have obscured the results [126]. A more recent study conducted in 2021 by Gholamigeravand et al. injected selenium nanoparticles together with adipose-derived MSCs in an AD mouse model. Compared with stem cell injection alone, they showed the combination therapy could effectively enhance cognitive function, as demonstrated in the novel object recognition (NOR) test. They also observed that amyloid-β deposition was also greatly reduced. They found the migration and survival of transplanted stem cells was enhanced in the presence of selenium nanoparticles. These studies demonstrate that stem cell-based therapies are effective and show promise as potential treatments for AD [129].

Although the etiology of AD is known to be multifactorial, genetic factors have been shown to play an important role in the disease’s development [130]. With advances in genetics, rats can now be genetically altered to mimic the AD pathology in humans. Genetically modified rats that can recapitulate AD pathology have been used to study AD pathogenesis and to test the effectiveness of treatments. A preclinical study by McGinley et al. in 2018 transplanted human neural stem cells in the fimbria fornix of an AD B6C3-Tg mouse model. They found significant cognitive improvements in NOR and MWM tests. Although the levels of cholinergic neurons and synapse-related proteins were unaffected, the levels of amyloid plaque were greatly reduced. However, this study did not provide sufficient details to validate the use of stem cell therapy in this preclinical model [131]. Another similar study by Kim et al. (2020) used human amniotic epithelial stem cells for implantation instead of human neural stem cells. They showed the stem cell transplantation was able to improve cognitive function in MWM and Y-maze tests. Furthermore, Congo red staining showed there were reduced levels of amyloid plaques in the mice brains. In addition, there was reduced activity of beta-secretase, which is involved in amyloid production [132,133]. A study by Losurdo et al. in 2020 administered MSC extracellular vesicles intranasally, which is a novel route of administration. They found the MSC-EVs could dampen the activation of microglia cells and increase dendritic spine density, suggesting neuroprotective effects in transgenic mice [134]. All these recent studies support the effectiveness of stem cell therapy on AD rodent models.

## 8. Clinical Studies of Stem Cell Implantation in Diseases Other Than AD

There have been many clinical studies of stem cell therapies for many kinds of disease, including spinal cord injury, Parkinson’s disease, and pancreatic beta cell insufficiency (diabetes mellitus). Although stem cell therapies show substantial promise, little success has been achieved in the treatment of many diseases. Five vital clinical studies have been chosen to evaluate the effectiveness of stem cells therapies in different diseases (Table 2).

A clinical trial on the use of human ES cells in congenital urea cycle disorder demonstrated therapeutic potential [145]. This was the world’s first clinical trial to use human ES cell-derived hepatocytes for the treatment of liver disease, which was verified to be both safe and effective. The patient, a 6-day-old baby with liver disease who was unable to detoxify ammonia, received ES cell injections into a blood vessel in the liver [145]. The implantation of ES cells was considered as a “bridge treatment” while the patient was waiting to receive a liver transplant 3 to 5 months after birth. After cell transplantation, there was no longer any increase in blood ammonia and the patient successfully completed the liver transplantation [145]. This study shows that stem cell therapy could be applied to prevent hyperammonemia while waiting for a liver transplantation. A study by Mendonça et al. investigated the effects of intralesional injection of bone marrow mesenchymal stem cells on spinal cord injury [146]. This Phase I trial observed inconsistent improvements in the sensitivity of all patients, with eight of them developing motor functioning in the lower limbs and seven of them improving their AIS. Although this study showed various improvements in patients, the sample size of 14 is too small to make a generalized conclusion [146]. A study on the effects of the umbilical cord blood-derived mesenchymal stem cells in patients with Parkinson’s disease was reviewed by Díaz in 2019 [147]. This ongoing study has not reported any outcome measurements or adverse effects so far, and the results need to be analyzed in the future. Other studies have shown that transducing stem cells with fibroblast growth factor-20 can increase their dopamine level to induce tyrosine carboxylase-positive cells [148].

A phase II study on the effects autologous bone marrow-derived stem cell transplantation on diabetes mellitus, reported by Bhansali et al., showed that the treatment ameliorated beta-cell functioning, as observed by a significant improvement in the glucagon-stimulated C-peptide levels and HOMA-B [149]. Moreover, the dosage requirement for insulin was also found to be decreased. There were no serious adverse effects, suggesting the stem cell treatment was safe, although the small study size is a concern and will require a larger study to verify the results [149]. A Phase I/II open-label study on the intrathecal transplantation of bone marrow-derived autologous mononuclear cells in 50 patients with Huntington’s disease showed improvement in cognitive and psychiatric symptoms, neuropsychiatric behaviors, writhing motions, abnormal posturing, and increased life expectancy [150]. Due to its large sample size, these results should be generalizable to the wider population of patients with Huntington’s disease [150]. As no information was given regarding any adverse reactions, its safety should be considered before clinical use.

## 9. Clinical Research on Stem Cell Therapies in AD

Although the concept of stem cell therapy has been around for more than 20 years, clinical trials investigating their effects in AD patients have been limited due to ethical issues, hindering their development as a treatment for AD [152]. Six clinical trials on stem cell-based therapies in AD are reviewed (Table 3).

The various cell therapies in the six clinical trials reviewed all used MSCs, which were derived from different tissues. The stem cells used in these clinical trials were mostly administered via the intravenous route, while stem cell transplantation in one trial was performed through surgical incision. The wide use of MSCs in many studies is due to their reported viability, safety, and efficacy [153]. Previous studies showed MSCs have tissue-regenerative properties, paracrine effects on the microenvironment, and immunomodulatory ability without causing immunological rejection in xeno-transplantation and allo-transplantation. As extensive research has been conducted on the efficacy and safety of MSCs, their use in clinical trials should accelerate the advancement of stem cell therapy for AD.

The first clinical trial investigating the effect of implanted stem cells in AD was conducted in 2011 by scientists at Medipost Co Ltd., Shandong, China [154]. This phase 1 trial focused on the safety and efficacy of using human umbilical cord blood-derived MSCs (hUC-MSC). They investigated dose-limiting toxicity of hUC-MSC injection in patients with mild-to-moderate AD. The clinical effectiveness of neural stem cells in treating AD patients was also not reported by Medipost Co Ltd. Moreover, several adverse effects were observed during the 12-week follow-up period. There were reports of acute adverse symptoms such as hearing discomfort, headaches, and dizziness, but there were no severe complications. This study also had several limitations including only nine patients and short follow-up period, and the design of this clinical trial was not an open-label trial. Such blinded studies can lead to the risk of placebo effects, as they are prone to psychogenic bias [155]. For example, patients with AD could be aware that they have received stem cell therapy, or their relatives could mention or ask them about their treatment, which could bias the experimental results.

In 2011, Ra et al. investigated the safety of human adipose tissue-derived mesenchymal stem cell infusion in both humans and animals [135]. The preclinical part of this study has already been discussed above. In their clinical trial, spinal cord injury patients between the ages of 23 and 54 with differing symptoms (seven quadriplegic and one paraplegic) were included in the study. There were no significant differences in the post-injection laboratory findings, physical examination, vital signs, and electrocardiogram, although there were some improvements in spinal cord injury after the neural stem cell treatment. They found the average area of spine damage was decreased from 134.50 ± 95.69 mm^2^ to 122.93 ± 99.45 mm^2^ after 12 weeks. Overall, 19 adverse events were observed including chest pain, chest tightness, and mild fever, although none of the patients developed any serious complications. Besides the adverse events from the stem cell therapy, the study sample was also too small, and the observational period was too short (26 weeks); additionally, they did not include AD patients [135]. Another clinical open-label study by Niu et al. with a longer follow-up of up to a year had a better study design and provided details of the protocols [156]. A more recent clinical trial by Oliva et al. [157] was a double-blind, randomized, placebo-controlled Phase I clinical trial. This study showed a newly developed stem cell, Longeveron allogenic mesenchymal stem cell, (LMSCs) was highly safe and tolerable. Recently, Liu et al. [20] reported two ongoing studies, one using umbilical cord-derived allogeneic hMSCs intravenously and the other using bone marrow stem cells (BMSCs) intravenously, intranasally, and with near-infrared light. These studies are currently recruiting, and their findings will need to be analyzed in the future. Although the use of stem cells in AD patients have shown relatively good results and are generally safe and tolerable, their efficacy in humans has yet to be established.

## 10. Mechanisms of Stem Cell Therapy in AD

In recent years, MSCs, ESCs, iPSCs, and brain-derived NSCs have been used in stem cell research in AD [20]. Embryonic stem cells are derived from the inner cell mass of pluripotent blastocysts [158]. They are pluripotent and can generate all cell types including mesodermal, endodermal, and ectodermal cells. However, their clinical application has been limited due to immune rejection and possible teratoma formation. Therefore, other types of stem cells such as MSCs have been used for stem cell therapy in AD [159]. A review paper by Liu et al. established the effectiveness of MSCs over ESCs in improving spatial learning and preventing memory decline [20]. On the contrary, Wang et al. showed that ESCs could improve mental aptitude in AD rodent models [116]; hence, the use of ESC in treating AD cannot be completely disregarded.

There is a growing number of studies showing the effectiveness of stem cell therapies in AD patients. However, the mechanisms underlying their effectiveness are not fully established. A study conducted by Zhu et al. in 2020 [144] showed that after engraftment of neural stem cells, some cells remained at the injection site, whereas some cells migrated to surrounding regions in the brain such as the corpus collosum. Cells were labeled with glial fibrillary acidic protein (GFAP) and doublecortin (DCX) to observe their differentiation into either astrocytes or other neurons. A small amount of the neural stem cells was observed to differentiate into astrocytes, whereas most differentiated into DCX+ neurons. These findings show that neural stem cells have the ability to both differentiate and migrate to facilitate repair of AD rodent brain. This finding was also supported in a study by Li et al., which showed stem cells could be differentiated into choline acetyltransferase (ChAT)-like neurons, supporting cognition [137]. Zhu et al. also reported that NSC transplantation increased levels of SYP, MAP-2, and SYP proteins. These proteins, especially MAP-2, are involved in microtubule assembly and are found at higher levels in dendrites [160]. The increased protein levels suggest that neural stem cells can increase neurogenesis. The NSC transplants also increased the level of ChAT neurotransmitters in the forebrain, indicating the NSCs also play a role in protecting cholinergic neurons. To investigate how NSCs can enhance neurogenesis, a study conducted by Se et al. examined the Wnt signaling pathway in a mouse model treated with NSCs [136]. They found MSC implantation increased the expression levels of GFAP, SOX2, Ki-67, HuD, and nestin, suggesting that NSCs can enhance neurogenesis possibly through the Wnt signaling pathway. Another study by Nakano et al. demonstrated that the implantation of bone marrow-derived MSCs in the hippocampus enhanced cognitive function through increasing the expression of microRNA-146a in AD rodent models [139]. To further investigate how the increased expression of microRNA-146a improved cognitive function, tissue sections were stained with synaptic marker (synaptophysin) to examine the synaptic density. They found AD rodents had increased synaptic density, confirming that microRNA-146a can induce synaptogenesis. As astrocytes were also reported to have a role in synaptogenesis [161], Nakano et al. further investigated the relationship between microRNA-146a, astrocytes, and synaptogenesis. They found that bone marrow-derived MSC exosomes containing miRNA-146a were taken up by astrocytes. They concluded that miRNA-146a was absorbed into astrocytes prompting synaptogenesis, leading to improved cognitive functions in AD rodents. However, how miRNA-146a is taken up by astrocytes leading to synaptogenesis is still unclear.

Stem cells were previously shown to alter the concentration of amyloid-β and tau, leading to cognitive improvements in AD. However, a study by Jones et al. [162] showed that stem cells increased synaptic density in the brain via BDNF, rather than directly altering amyloid-β and tau levels. These findings support the studies by Nakano et al. and Ager et al. [143]. Coming full circle, recent studies have reestablished support for the previous view that stem cells can reduce both amyloid-β and tau levels, leading to enhanced cognitive function. Recent studies in the past 5 years from Chen et al., Li et al., Lee et al., McGinley et al., and Lu et al. showed that MSC implantation could reduce levels of amyloid-β, leading to improved cognition in mice [131,138,140,142]. In addition to stem cells modulating BACE1 expression to alter APP processing, Lee et al. also showed that stem cell transplantation could inhibit tau phosphorylation. They found that stem cells reduced tau phosphorylation through the activation of the Trk-dependent Akt/GSK3β signaling pathway [142].

A study by Lu et al. suggested stem cell implantation could reduce amyloid-β 40 and 42 accumulation by a clearance mechanism involving increased IDE and NEP levels [140]. A study by Chen et al. showed stem cells prevented the downregulation of BDNF, which further supports the studies by Jones et al., Nakano et al., and Ager et al. [138,139,143,162]. Neuronal memory and synaptic plasticity-related genes, such as BDNF, SYP, GluR2, and GRIN2B, have been shown to improve memory. Moreover, Chen also showed that MSC-exosomes could improve brain glucose metabolism as measured by [18F] FDG-PET. Abnormal expression of HDAC4 was decreased after the treatment, whereas the expressions of HDAC4 target genes like Homer protein homolog 1, synapsin II, and leucine-rich glioma inactivated 1 were increased, leading to amelioration of learning and memory deficits [138].

Lu et al. also reported the effects of stem cell treatment in AD involving neuroinflammation and pericytes [140]. They discovered that hNSC transplantation could reduce the density of astrocytes and microglia, suggesting the inhibition of neuroinflammation leading to improved cognition. A study by McGinley et al. also supported the modulation of microglial activation in vivo [68]. Lu et al. found that hNSC transplantation enhanced endogenous neurogenesis, while reducing pericytic and synaptic loss in the hippocampus of APP/PS1 mice. They suggested the increase in pericytes enhanced amyloid-β clearance by facilitating phagocytosis and transport out of the CNS across the BBB, while maintaining BBB permeability, oxygen supply, and metabolites in collaboration with endothelial cells. Pericytes were also found to modulate neurogenesis and angiogenesis, with corresponding changes in cognition. Last but not the least, Li et al. revealed that hNSC transplantation elevated NAA and Glu levels and lowered choline and myo-inositol levels, leading to improved neuronal activity (Figure 1).

## 11. Limitations and Future Perspectives

As the research on stem cells has progressed, several limitations on the use of stem cells in AD have been revealed. In 1988, Date et al. [163] explored the issue behind the immune rejection of stem cells when transplanted into the body. Although they did not specifically study AD rodent models, they found that NSC grafts resulted in class 1 MHC antigens on the newly developed neuronal tissue. They also observed increased numbers of reactive astrocytes in the graft, indicating immune rejection activities. In earlier studies [116], it was found that after implantation of ESCs, the grafted cells developed into a teratoma. Subsequently, other types of stem cells such as MSCs have been used, which have a lower chance of developing into a teratoma. Although the stem cells developed by Medipost Ltd. Co. were reported to be safe against immune rejection, future studies will be needed to confirm their safety in humans. Several ethical concerns need to be considered when using stem cells [164]. For example, it can be difficult to obtain permission to treat infants with embryonic tissues such as stem cells. Another concern by Tan et al. is that the authenticity of the patient may be threatened after stem cell implantation, as they may believe the treatment might enhance or change the retention of particular memories [165]. In the brain, the complex neural circuits play a role in so-called higher brain functions. The transplanted stem cells have the potential to not only decrease neural damage, but also to alter neural circuits. The treatment of patients with mild or severe Alzheimer’s disease may lead to completely different personalities through the generation of new neural pathways. Tan et al. also suggested that some unwanted consequences may occur, such as the overgeneralization of enhanced memories that could lead to anxiety disorders. A study by Aguila et al. indicated another limitation could be the complexity of the cellular programming and reprogramming of stem cells. For example, the functionality of stem cells might be affected by intrinsic factors or by abnormal extrinsic factors like Wnt1, Fgf8, and retinoic acid [166]. Therefore, researchers need to conduct more studies to ensure the intact functionality of the transplanted stem cells.

## 12. Conclusions

Alzheimer’s disease affects millions of individuals globally every year and is becoming an increasing health burden. Although various therapeutic treatments for AD are available such as cholinesterase inhibitors, memantine, NMDA receptor antagonists, and antioxidants, their effectiveness are low. Stem cell therapy as a potential treatment for AD has sparked the interest of many scientists. Research on human neural stem cells has greatly increased in recent years, demonstrating their potential beneficial effects. Over the years, many preclinical studies have been conducted to investigate the effectiveness of stem cells in AD. Preclinical studies of stem cell treatments in rodent AD models have successfully shown they can improve cognitive functions. However, more clinical trials are needed to establish their effectiveness and safety in humans. As the main goal of AD treatments is to cure AD or slow down disease progression, more research is needed to maximize the effectiveness of stem cells. Even if stem cells are proven to be useful in combating AD, there are also hurdles in generalizing this therapy due to their prohibitive cost.

This review has some weaknesses as only a few highly relevant studies were included, and the scope of the study did not allow us to comprehensively compare all trials. Moreover, the review does not present a full analysis of all of stem cell developments in AD, although this review includes many up-to-date studies. Considering the reviewed studies, we can foresee more research will emerge in this field, providing useful insights on the administration route, combination therapies, and human applications in the future.

## Figures and Tables

**Figure 1 ijms-22-10151-f001:**
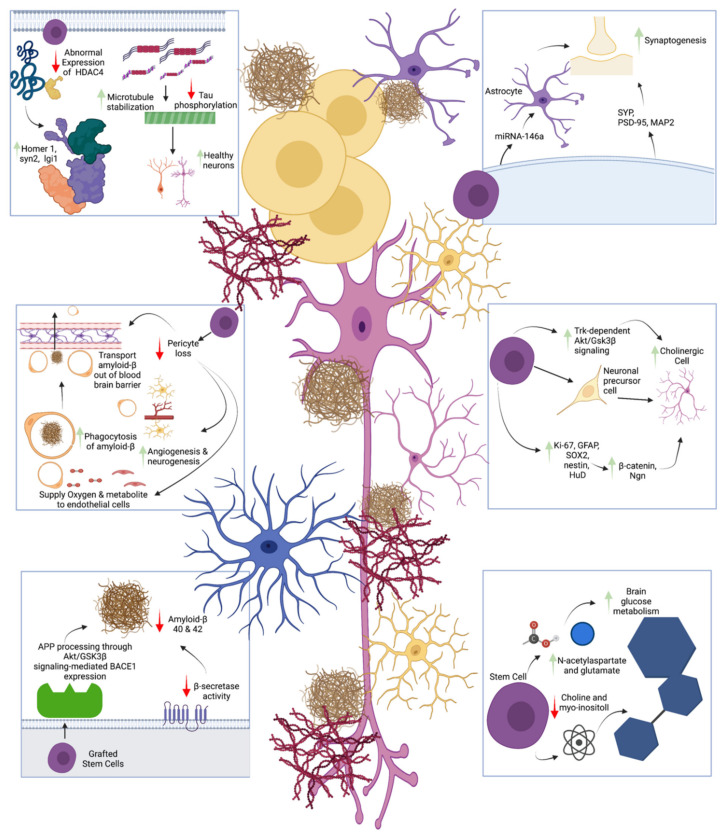
Mechanism of stem cell amelioration of cognition. Stem cells enhance formation of Syp, PSD-95, and MAP2, which can stimulate synaptogenesis. miRNA-146a is also secreted by stem cells and later taken up by astrocytes, leading to synaptogenesis. Stem cells can increase cholinergic neuron production in three ways: 1. increased Trk-dependent Akt/Gsk3β signaling, 2. differentiation into cholinergic cells, and 3. increased secretion of Ki-67, GFAP, SOX2, nestin, and HuD; increased secretion of β-catenin and neurogenin (Ngn); and increased production of cholinergic cells. Brain glucose metabolism is also increased after stem cell implantation through increased N-acetylaspartate (NAA) and glutamate secretion and decreased choline and myo-inositol secretion. Amyloid-β 40 & 42 are known to cause AD. Stem cells can alter APP processing through Akt/GSK3β signaling-mediated BACE1 expression and decrease β-secretase activity to decrease amyloid β concentration and improve cognition. Pericytes also have a role in AD. Transplantation of stem cells can inhibit pericyte loss to: 1. promote phagocytosis of amyloid-β, 2. transport amyloid-β across the BBB, 3. supply oxygen and metabolites to endothelial cells, and 4. increase angiogenesis and neurogenesis, which improves cognitive function. Stem cells have a potent effect on tau phosphorylation in AD. They decrease tau phosphorylation, leading to increased microtubule stabilization and healthier neurons, thereby ameliorating cognition [141].

**Table 1 ijms-22-10151-t001:** Preclinical studies investigating the effectiveness or safety of stem cell therapies in AD animal models.

Authors	Models and Transgene	Dosage and Injection Site	Type of Stem Cells	Behavioral Outcome	Mechanisms & Physiological Effects
Farshad et al., 2009 [124]	Sprague–Dawley rats with nucleus basalis of Meynert lesion	One surgery injection into coordinates from bregma: AP = −0.9, L = 2.8, V = 6.8 with 2 × 10^5^ cells in 2 μL.	Embryonic stem cells	Learning, memory, and spatial cognition improved in the MWM test	Successful in vitro differentiation of ESC to neuronal precursor cells.
Liu et al., 2013 [126]	p75-saporin lesion models (8 to 10 weeks old)	One surgery injection into ventricle with about 100,000 cells	hESCs (line H9, passages 18–35; line H1 passages 30–36)	Learning, memory, and spatial cognition improved in the MWM test	hESCs differentiated into primitive neuroepithelia, MGE-like progenitors, then into cholinergic spinal motor neurons in vitro. MGE progenitors efficiently induced by SHH, and ISL1, OLIG2, and ASCL1 (expressed in lateral ganglionic eminence (LGE) and MGE cells in vivo). Grafted neural progenitor cells produced neurons and glia.
Ra et al., 2011 [135]	SCID mice (91 males and 91 females, 6 weeks old)	One injection of different MSC dosages in the tail vein: Saline control Low dose (5 × 10^6^ hAdMSCs/kg B.W.) Medium dose (3.5 × 10^7^ hAdMSCs/kg B.W.) High dose (2.5 × 10^8^ hAdMSCs/kg B.W.)	MSCs	No abnormal side effects	No specific pathological changes were observed in mice organs including lungs. No significant changes in the hematology, urine, ophthalmic test, and blood chemistry results.
Se et al., 2015 [136]	Male C57BL/6 mice receiving Aβ1—42 injection (6 weeks old, not transgenic)	One intracerebroventricular injection of 1.0 × 10^6^ cells/mouse	MSCs	Behavioral analysis of the radial Arm Maze test showed the MSC treatment significantly improved cognitive memory performance	Short-term treatment with MSCs in Aβ-treated NPCs slightly increased expression of proliferation marker Ki-67 and neuronal progenitor markers GFAP, SOX2, and nestin without changing HuD expression. Long-term treatment significantly increased expression of the proliferation marker, neuronal progenitor markers, and the neuronal marker HuD compared to Aβ treatment alone. In Aβ-treated NPCs co-cultured with MSCs, long-term MSC treatment significantly increased expression of β-catenin and Ngn1 compared to Aβ treatment alone, showing Wnt/β-catenin signaling is involved in MSC-induced increased survival and neuronal differentiation in NPCs. MSCs enhanced expression of β-catenin and Ngn1 in AD animal models. Double-stained BrdU and Ngn1 cells in the hippocampus were frequently observed in MSC-treated AD animals.
Li et al., 2007 [137]	Sprague–Dawley rats with synthetic Aβ 1–40 amyloid protein injected	One injection with 2–3 × 10^5^ cells into the hippocampus	BMSCs	Cognition improved in the MWM	Transplanted cells expressing NGF survived and differentiated into ChAT-like neurons.
Gholamigeravand et al., 2021 [129]	Adult male Wistar rats with streptozotocin-induced memory impairment	Five groups: Intact control group. Streptozotocin-injected group (STZ) received intracerebroventricular (ICV, 3 mg/kg) injection of streptozotocin in two half-doses on days 1 and 3. STZ group received single ICV transplantation dose of AMSCs (STZ + AMSC) 1 month after induction of the model. STZ-treated rats received daily oral administration of Se-NPs 0.4 mg/kg for 1 month (STZ + SeNP) via oral gavage feeding tube. STZ-injected rats received both AMSCs and SeNPs (STZ + AMSC + SeNP).	Adipose-derived mesenchymal stem cells (AMSCs)	NOR test showed cognitive improvements	Synergistic effects of SeNPs and AMSCs on memory function in STZ-injected rats with decreased Aβ deposition. Administration of SeNPs enhanced migration, survival, and BDNF hippocampus concentration of transplanted AMSCs rats.
Losurdo et al., 2020 [134]	Triple-transgenic 3xTg mice	PBS solution or extracellular vesicles in ∼5 μL spurts per nostril	MSCs	Not available	MSC-EVs decreased microglia activation in 3xTg AD mice. MSC-EVs increased dendritic spine density in 3xTg mice.
Kim et al., 2020 [133]	Tg2576 mice expressing mutant human APP containing the Swedish (K670N/M671L) mutation	Not given	hAESCs	Spatial learning and memory were improved in MWM test	Transplantation of hAESCs reduced amyloid burden.
Chen et al., 2021 [138]	J20 transgenic mouse model of C57BL/6 type (JAX-006293) and age-matched control group	One intravenous injection of 50 μg of purified MSC-exosomes (number not specified)	MSCsh	Cognitive function improved in the NOR test	MSC-exosomes decreased Aβ levels. MSC-exosomes prevented the downregulation of neuronal memory/synaptic plasticity-related genes such as Bdnf IV, SYP, GluR2, and GRIN2B. MSC-exosomes improved brain glucose metabolism as shown in the [18f] FDG-PET. MSC-exosomes inhibited astrocyte activation, upregulated neuronal memory, and synapse-related genes; levels of bdnfiv, syp, and glur1 were increased. HDAC4 expression decreased after treatment, and expressions of HDAC4 target genes Homer1 (Homer protein homolog 1), Syn2 (Synapsin II), and Lgi1 (leucine-rich glioma inactivated 1) were elevated.
Masako et al., 2020 [139]	Protein/presenilin 1 (APP/PS1) mice	Two times at a 2-week interval with 1 × 10^5^ BM-MSCs per mouse via intracerebroventricular injection	BM-MSCs	Improved learning and memory impairment shown by the probe test and hidden platform MWM	BM-MSCs decreased glial fibrillary acidic protein (GFAP)- and tumor necrosis factor (TNF) α-positive areas in astrocytes. BM-MSCs decreased M1 type activated microglia and increased M2 type activated microglia in AD model mice. MSC-CM secreted C-X-C Motif Chemokine Ligand 5 (CXCL5), monocyte chemoattractant protein-1 (MCP-1), beta-nerve growth factor (β-NGF), tissue inhibitor of metalloproteinase-1 (TIMP-1), and vascular endothelial growth factor-A (VEGF-A). The numbers of F4/80-positive macrophages and the intensity of transthyretin (TTR) were increased in CP. The expression of miR-146a in the hippocampus was significantly upregulated. Expression of TRAF6 in the subiculum area was significantly decreased. BM-MSC-derived exosomes suppressed the expression of NF-κB by transferring miR-146a into astrocytes. BM-MSC-derived exosomal miR-146a were taken up into astrocytes leading to anti-inflammatory effects.
Lu et al., 2021 [140]	APP/PS1 double transgenic mice expressing mutant human amyloid precursor protein (APPswe) and presenilin 1 (PS1-dE9) under a mouse prion protein promoter	One injection with 8 μL (1 × 10^6^, 4 μL/side) of hNSCs or saline on both sides of the nasal cavity	hNSCs	Cognition was improved in NOR and MWM tests	hNSCs attenuated Aβ40 and Aβ42 accumulation in APP/PS1 mice and promoted clearance via increased IDE and NEP levels. hNSC transplantation reduced the density of astrocytes and microglia, indicating an inhibition of neuroinflammation. Intranasal transplantation of hNSCs enhanced endogenous neurogenesis, ameliorated pericytes, and synaptic loss in the hippocampus of APP/PS1 mice. Increased number of pericytes increased Aβ clearance by phagocytosis and transportation of Aβ out of the CNS through the BBB, maintaining BBB permeability, and supplying oxygen and metabolites in endothelial cells. Increased number of pericytes modulated neurogenesis and angiogenesis correlated with cognition.
Li et al., 2016 [141]	APP/PS1 Tg mice	A Hamilton micro syringe fixed on the stereotaxic apparatus was inserted 2.5 mm under the dura, and a 4 μL NSC suspension (at 1 × 10^5^/μL) or PBS was injected into the brain gently	hNSCs	Alleviated cognitive, learning, and memory but not anxiety deficits in NOR and MWM tests	Transplantation of hNSC reduced soluble Aβs, but not insoluble Aβs and plaque burden in AD mice brains. Transplantation of hNSC rescued neuronal loss and connectivity in AD mice brains. Improved neuronal metabolic activity, elevated NAA and Glu peaks, but lowered choline and myo-inositol.
Lee et al., 2015 [142]	NSE/APPswe transgenic mice	One injection of 5 μL of vehicle or hNSC suspension 1 × 10^5^ cells/μL bilaterally into lateral ventricles (LVs; 0.1 mm caudal, 0.9 mm bilateral to bregma, and 2.0 mm ventral from the dura mater)	hNSCs	Spatial memory was improved in the MWM	Human NSC transplantation inhibited tau phosphorylation, activated Trk-dependent Akt/GSK3β signaling, and reduced Aβ42 levels. Human NSC transplantation altered APP processing by modulating BACE1 expression, and decreased astrogliosis and microgliosis. Human NSC transplantation attenuated microglial activation through cell-to-cell contact and secretory molecules. Transplantation of hNSC reduced soluble Aβs, but not insoluble Aβs and plaque burden, in AD mice brains.
McGinley et al., 2018 [131]	Male B6C3-Tg (APPswe/PSEN1ΔE9) 85Dbo/J (APP/PS1; n = 20) mice (stock #034829-JAX; Jackson Laboratory, Bar Harbor, ME)	Unknown number of stem cell injected at the following coordinates (bregma/lateral/ventral): −0.82/0.75/2.5, −1.46/2.3/2.9, −1.94/2.8/2.9 mm	NSCs	NSCs targeted to the fimbria fornix improved cognitive function in NOR and MWM tests	Transient NSC engraftment reduced Aβ plaque pathology. NSCs modulated microglial activation in vivo and in vitro.
Ager et al., 2018 [143]	APP/PS1 AD mouse model (8 weeks old)	2-week interval with 3 × 10^5^ cells per hippocampus	Unique line of human cortex-derived neural stem cells (NSCs; NSI-HK532-IGF-1)	3xTg-AD-HuCNS-SC mice performed significantly better during the probe trial in MWM Significantly enhanced place-dependent memory performance with transplanted stem cell in the NOR task	Substantial survival of transplanted human cells in two distinct immunosuppressed transgenic models of AD-associated neurodegeneration. Increased synaptophysin (SYP), synapsin, and growth-associated protein-43 (GAP-43) in mice, indicating increased synaptic density.
Zhu et al., 2020 [144]	APP/PS1 (APPswe, PSEN1dE9) double transgenic mice (5 months old)	One injection of 5 µL × (1 × 10^5^ cells/µL) in the hippocampus bilaterally	NSCs derived from the embryonic brain (E12.5–14.5 days) of pregnant EGFP-labeled mice	Improved spatial learning and memory ability in the MWM	Engrafted stem cells survived and partly remained at the injection site. Some engrafted stem cells migrated to surrounding regions including the corpus callosum and adjacent cortex. Some engrafted stem cells experienced morphologic changes and differentiated into GFAP+ and DCX+ cells, i.e., astrocytes. SYP, PSD-95 (postsynaptic density protein 95), and MAP-2 proteins significantly increased after stem cell implantation. Neural stem cell transplantation increased ChAT protein levels in the basal forebrain.

Abbreviations: hNSCs, human central nervous system stem cells; MWM, Morris water maze; GFAP, glial fibrillary acidic protein; DCX, doublecortin; SYP, synaptophysin; MAP-2, microtubule-associated protein 2; ChAT, choline acetyltransferase; SCID, severe combined immunodeficient; hAdMSCs, human adipose-derived mesenchymal stem cells; Ngn1, neurogenin 1; BrdU, bromodeoxyuridine; hESCs, human embryonic stem cells; SHH, Sonic hedgehog; ISL1, insulin gene enhancer protein 1; OLIG2, oligodendrocyte transcription factor; ASCL1, achaete-scute family BHLH transcription factor 1; BMSCs, bone marrow-derived stem cells; NGF, nerve growth factor; BdnfIV, brain-derived neurotrophic factor IV; GRIN2B, glutamate ionotropic receptor NMDA type subunit 2B; NEP, neprilysin; IDE, insulin-degrading enzyme; NOR, novel object recognition test.

**Table 2 ijms-22-10151-t002:** Present clinical studies of stem cell therapies in diseases.

Authors	Disease, Clinical Phase, and Duration	Study Design	Type of Stem Cells and Implantation Route	Dosage and Concentration	Outcome Measures	Clinical Evaluation	Adverse Effects
“Clinical trial with human...” [145]	Congenital Urea Cycle Disorder Phase N.A.	6-day-old baby	Human embryonic stem (HES) cell-derived hepatocytes Intravenous infusion	N.A.	Blood ammonia concentration	Liver transplantation	No complications from the surgical procedure.
Mendonça et al., 2014 [146]	Spinal Cord Injury Phase 1 6 months	Open-label, single-group assignment 14 patients (18 to 50 years old)	Bone marrow mesenchymal stem cells (BMMSC) Intralesional injection	5 × 10^6^ cells/cm3 single dose	Feasibility and Safety of BMMSC, Functional Improvement in muscle strength and sphincter control	Frankel Scale American Spinal Injury Association Impairment Scale (AIS) Somatosensory evoked potential (SEP)	One patient developed cerebrospinal fluid (CSF) leak due to intervention practices. No severe side effects or other complications.
Díaz, 2019 [151]	Parkinson’s Disease Phase 1 1 year	Open-label, single-group assignment 20 participants (40 to 60 years old)	Umbilical cord blood-derived mesenchymal stem cells (UC-MSCs) Intravenous infusion	10–20 million cells once a week for 3 weeks	N.A.	Unified Parkinson’s Disease Rating Scale (UPDRS) MMSE Hoehn and Yahr staging (H-Y) Hamilton Depression Scale 24 (HAMD 24) Hamilton Anxiety Scale 14 (HAMA-14) Adverse reaction	N.A.
Bhansali et al., 2009 [149]	Diabetes Mellitus Phase II 6 months	Open-label, single-group assignment 10 patients (30 to 75 years old)	Autologous bone marrow-derived stem cells (ABMSCs) Angiographically	N.A.	Reduction of insulin requirement by > 50% Increment in glucagon stimulated C-peptide levels Reduction of insulin requirement Improvement of HbA1c levels compared to baseline	Glucagon stimulated C-peptide levels Insulin dosage HbA1c levels	No serious adverse effects were noted.
Rodrigues & Edward, 2018 [150]	Huntington’s Disease Phase I/II 36 months	Open-label, single-group assignment 50 participants (35 to 44 years old)	Bone marrow-derived autologous mononuclear cells (BMAMNCs) Intrathecal transplantation	100 million stem cells per dose	Improvement in cognitive and psychiatric symptoms Improvement in neuropsychiatric behavior Increase in life expectancy Improvement in writhing motions or abnormal posturing Improvement in compulsive behavior	N.A.	N.A.

Current clinical studies of stem cell therapies in diseases other than AD. Abbreviations: human embryonic stem cells; BMMSCs, bone marrow mesenchymal stem cells; AIS, American Spinal Injury Association Impairment Scale; SSEP, somatosensory evoked potential; CSF, cerebral spinal fluid; UC-MSCs, umbilical cord blood-derived mesenchymal stem cells; UPDRS, Unified Parkinson’s Disease Rating Scale; MMSE, Mini Mental State Examination; H-Y staging, Hoehn and Yahr staging; HAMD 24, Hamilton Depression Scales 24; HAMA-14, Hamilton Anxiety Scale 14; ABMSCs, autologous bone marrow-derived stem cells; HbA1c, glycated hemoglobin; BMAMNCs, bone marrow-derived autologous mononuclear cells; N.A., not available.

**Table 3 ijms-22-10151-t003:** Clinical studies investigating the effectiveness or safety of stem cell therapies in AD patients.

Authors	Clinical Phase & Duration	Study Design	Type of Stem Cells and Implantation Route	Dosage and Concentration	Outcome Measures	Clinical Evaluation	Adverse Effects
Liu et al. [20]	Phase I active 65 weeks	Open-label, prospective, single-group assignment 50 to 85 years old	Umbilical cord-derived, allogeneic hMSCs Intravenous infusion	100 million cells per infusion	N.A.	Adverse events evaluation ADAS-Cog MMSE Geriatric Depression Scale (GDS) Odor identification test Alzheimer’s Disease Related Quality of Life (ADRQL-40) Alzheimer’s Disease Cooperative Study Activities of Daily Living (ADCS-ADL) Neuropsychiatric Inventory-Q (NPI-Q)	N.A.
Oliva et al., 2019 [157]	Phase I completed Active for Phase II 1 year	Randomized controlled trial 50 to 80 years old	Longeveron MSCs Intravenous infusion	Once for 20 million LMSCs (low-dose), 100 million LMSCs (high-dose), or placebo	N.A.	Cognitive assessments Patient-reported outcomes (PROs) Biomarkers (serum, CSF, and MRI).	No serious adverse events
Ra et al., 2011 [135]	Phase 1 7 months	Single group 8 male patients (19 to 60 years old)	Autologous adipose-derived MSCs Intravenous infusion	4 × 10^8^ autologous hAdMSCs per patient	Improvement in some of the cases	Blood chemistry, HBV/HCV, hematology, and urinalysis, and were screened for HIV, and VDRL. A pulmonary function test, chest X-ray, spinal cord independence measure (SCIM), visual analog scale, electrophysiological examination of motor, spinal magnetic resonance imaging, somatosensory evoked potentials (MEP and SEP, respectively), and neurological examinations using ASIA were obtained for each patient.	19 adverse events were observed in 8 patients, including chest tightness, chest pain, and mild fever.
Niu et al., 2016 [156]	Phase I/II Active 1 year	Open-label, self-control, single-center prospective trial 30 patients (50 to 85 years old)	hUC-MSCs Intravenous into the median cubital vein	0.5 × 10^6^ hUC-MSCs/kg	N.A.	Adverse effects evaluation AD improvement, e.g., CIBIC, MMSE, ADL, and NPI	N.A.
Medipost Co Ltd. [154]	Phase I N.A.	Open-label, single-group assignment 9 patients (50 to 75 years old)	Human umbilical cord blood-derived MSCs	Dose A—250,000 cells per 5 µL per entry site, 3 million cells per brain Dose B—500,000 cells per 5 µL per entry site, 6 million cells per brain	N.A.	Adverse event evaluation Changes in ADAS-cog	N.A.
Liu et al. [20]	Phase N.A. 12 months	Open-label, non-randomized, parallel assignment 18 years and older	Bone marrow stem cells (BMSCs) Intravenous Intranasal topical Near-infrared light	14 cc of BMSC fraction 1 cc of BMSC fraction 14 cc of BMSC fraction	N.A.	MMSE Autism Spectrum Quotient Exam (AQ) ADCS-ADL	N.A.

Clinical studies investigating the effectiveness or safety of stem cell therapies in AD patients. Abbreviations: hUC-MSCs, human umbilical cord blood-derived mesenchymal stem cells; ADAS-Cog, AD Assessment Scale-cognitive Subscale; S-IADL, Seoul Instrumental Activities of Daily Living; MMSE, Mini Mental State Examination; hMSCs, human mesenchymal stem cells; GDS, Geriatric Depression Scale; ADRQL-40, Alzheimer’s Disease Related Quality of Life; ADCS-ADL, AD Cooperative Study Activities of Daily Living; NPI-Q, Neuropsychiatric Inventory-Q; LMSCs, Longeveron mesenchymal stem cells; PROs, patient-reported outcomes; CSF, cerebral spinal fluid; MRI, magnetic resonance imaging; hAdMSCs, human adipose-derived mesenchymal stem cells; HBV, hepatitis B virus; HCV, hepatitis C virus; HIV, human immunodeficiency virus; VRDL, venereal disease research laboratory test; SCIM, spinal cord independence measure; MEP, motor evoked potential; SEP, sensory evoked potential; ASIA, American Spinal Injury Association; CIBIC, Clinician Interview-Based Impression of Change; ADL, Activities of Daily Living; NPI, Neuropsychiatric Inventory; BMSCs, bone marrow stem cells; hESCs, human embryonic stem cells; AQ, Autism Spectrum Quotient Exam; N.A., not available.

## Data Availability

Not applicable.

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
