# Peer review of "Therapeutic Potential of Human Stem Cell Implantation in Alzheimer’s Disease"

_ijms, 2021, doi:10.3390/ijms221810151_

Round 1
Reviewer 1 Report
Hau Jun Chan et al. MS# ijms-1356132
Therapeutic potential of human stem cell implantation and neurogenesis in Alzheimer’s disease
In this manuscript, the authors review published literature on the pre-clinical and clinical use of human stem cells in the context of Alzheimer’s disease (AD). The introduction and current treatments sections (refs. 1-29) are very general, but do provide useful background on the pathological hallmarks and proposed pathogenic mechanisms underlying the disease, as well as on the currently available therapies. In contrast, I found the remaining sections extremely superficial and confusing, and overall lacking scientific rigor. Studies on completely different types of stem cells are presented together for no apparent reason (other than being ‘stem cells’). The section ‘Therapeutic potential of neural stem cells’ (refs. 30-39) mainly reviews pre-clinical studies on PSC-derived NPCs, rather than on actual NSCs. The next section, ‘Preclinical research on neural stem cell therapies in AD’ (refs. 40-48) cherry-picks 9 papers on use of ‘stem cells’ in rodent models of AD, presenting data from 2 papers on PSC-derived NPCs, 1 paper on fetal NSCs, 1 paper on MSCs, 2 more papers on PSC-derived NPCs, 1 more paper on NSCs, 1 paper on ‘human amniotic epithelial stem cells’, and 1 more final paper on MSCs. The reasons for selecting these 9 papers in particular, and for jumping from one type of stem cell to the next and then back, are utterly unclear to me.
The next section (‘Clinical research on neural stem cell therapies in AD’, refs. 49-54), briefly summarizes the findings of 7 early-phase clinical trials of MSC-based cell therapy in AD. For unknown reasons, these cells are referred to as NSCs several times within this section. As a matter of fact, the authors conclude the section stating ‘Although the use of neural stem cells in AD patients have shown relatively good results, and are safe and tolerable, their efficacy in humans has yet to be established.’ Moreover, two of the alleged trials (refs 50 and 52) appear to refer to the same one.
The final sections of the manuscript (‘Mechanisms of human stem cell therapy in AD’ and ‘Limitations and future perspectives’), again mix information pertaining to different types of stem/progenitor cells in manners that create confusion in the reader. The conclusion then refers only to NSCs further contributing to the overall confusion.
Reviewer 2 Report
20 August 2021
Review on the manuscript titled “Therapeutic potential of human stem cell implantation and neurogenesis in Alzheimer’s disease” by Chan HJ et al., submitted to International Journal of Molecular Sciences (IJMS).
Manuscript ID: ijms-1356132
Dear Authors,
Alzheimer’s disease (AD) is the most prevalent neurodegenerative disease that causes dementia, but no effective treatment is established. The authors review stem cell therapy for the treatment of AD in preclinical and clinical studies. The stem cell therapy is effective and safe in animal studies, but few clinical data has been presented. Although there are limitations to overcome, the authors concluded that the stem cell therapy is a potential treatment for AD.
Please reconsider the following parts:
- Please use the template for IJMS.
- A graphical abstract is highly recommended.
- Pages 2, Abstract: Please expand the abstract to 200-220 words and clearly describe animal, preclinical, and clinical studies, including positive results in preclinical studies and the trial phases of clinical studies.
- Page 2, Keywords: Please list more keywords up to ten.
- Pages 3,4, Introduction, Paragraph 1: The term dementia is outdated. Please introduce the term neurocognitive disorder. The symptoms and comorbidity of depression and anxiety in dementia were discussed recently. Suggested references: https://doi.org/10.1007/978-3-319-56015-1_444-1; https://doi.org/10.3390/biomedicines9050517; https://doi.org/10.3390/jcm10091809.
- Page 3,4, Introduction, Paragraph 2: Please update the references. The pathogenesis of AD and neurodegenerative diseases is reviewed recently. Environmental factors may include even microorganisms and their toxins. Suggested reference: https://doi.org/10.3390/ijms21072431; https://doi.org/10.3390/sci3010016; https://doi.org/10.3390/ijms22168726.
- Page 5, Outline of the Review: If the authors intend the manuscript o be a systematic review, please present inclusion and exclusion criteria and PRISMA flow diagram besides the search terms.
- Pages 5,6, Current treatments for Alzheimer’s disease: Novel pharmacological approaches for dementia including repurposed drugs are reviewed recently. Suggested reference: https://doi.org/10.1007/978-3-319-56015-1_444-1.
- Pages 8-11, Preclinical research and Table 1: Please rearrange the content according to Research Domain Criteria (RDoC), translational studies to animal models, transgenic models of AD, and behavioral tests according to the constructs of RDoC. Suggested reference: https://www.nimh.nih.gov/about/advisory-boards-and-groups/namhc/reports/behavioral-assessment-methods-for-rdoc-constructs.
- Pages 11, Clinical research: Please summarize the methods including types of cells, administration route, outcome assessment, and possibly their rationales in the beginning of the section.
- Pages 13-17, Mechanisms: Please present a figure summarizing the section.
- Page 14, Conclusion: Please summarize the key findings, weaknesses in the review, potentials, the ultimate goal, research or knowledge needed to achieve, the biggest challenge in this goal, and future research directions, among others.
- Pages 14-16, references: Please cite more references at least more than 150 for review articles.
The manuscript contains one figure, two tables, and 75 references. The manuscript carries important value presenting the stem cell therapy of preclinical and clinical studies for the treatment of AD. However, the manuscript deserves to be revised as suggested above to improve the quality of the presentation. I reconsider this manuscript for publication after major revision.
I declare no conflict of interest regarding this manuscript.
Best regards,
Masaru Tanaka, M.D., Ph.D.
Reviewer 3 Report
ijms-1356132(Review): Therapeutic potential of human stem cell implantation and neurogenesis in Alzheimer's disease
Since the clinical application of human stem cell implantation is far away from the general clinical stage in any kind of disease, the theme of this review may be said to be too early for the fruitful discussions. However, the summary of the present literature search is informative to understand the present status of this area. Hence this manuscript is worth being published in the Journal. The review is well described and almost ready for the publication, The following suggestions may help to increase the value of this review.
Although this is a matter of the Editorial Office, the manuscript PDF file provided has no page numbers and no line numbers. It is very hard to pinpoint the lines mentioned below.
<Minor Points>
(1) The Introduction section should be improved. At the end of the Introduction, the current view of the etiology of Alzheimer's disease (AD) is briefly mentioned. Then suddenly it was followed by the method of PubMed database search. One suggestion is as follows.
1. The Introduction is ended in the last line of PDF Page 3 (patients following diagnosis can be up to 9 years [6].)
2. Then the new section of "Etiology of Alzheimer's disease" starts.
3. Next is " Current treatments for Alzheimer’s disease".
4. Next is " Therapeutic potential of neural stem cells", the title should be changed to, for example, "The alternative strategy for the treatment of Alzheimer’s disease".
5. "Due to the low efficacy of current treatments, pharmaceutical companies and medical institutes have been actively seeking alternative therapies for AD including neural stem cell (NSC) transplantation" (PDF page 6) is followed by " Here, we review and discuss preclinical studies and clinical trials of stem cell therapies for AD in relation to their efficacy and safety in AD patients. The mechanisms, therapeutic potential, and limitations of the stem cell therapies are fully discussed." (PDF page 5).
6. " Outline of the Review" section is moved here.
7. " Therapeutic potential of neural stem cells" section starts from "Rosenberg (1988) was first to successful(ly) graft genetically modified cells in a damaged brain to protect cholinergic cells from dying [32]."(PDF page 6).
(2) It is said as "It has been observed that all forms of AD share an increased production and decreased clearance of amyloid beta peptides."(PDF page 4). However, there are some exceptions as "arctic" mutation (https://www.alzforum.org/mutations/app-e693g-arctic) and Osaka mutation (https://www.alzforum.org/mutations/app-e693del-osaka) with little increase in Abeta but increase in protofibrils and aggregations, respectively. Hence the above sentence is not completely correct. Since the definition of AD is a dementia with senile plaques and tangles, it can be said that all the forms of AD have senile plaques and almost all of them share an increased production and decreased clearance of amyloid beta peptides.
(3) In the "Outline of the Review", the date of PubMed database search should be indicated. The numbers of the published articles in each year and the citation analysis may be interesting.
(4) Before the section "Clinical research on neural stem cell therapies in AD", there should be the section of the present clinical applications of human stem cell implantation in diseases other than Alzheimer disease. The difficulty of stem cell therapy is not limited to Alzheimer's disease. The idea of stem cell therapy is great but little achievement over many kinds of disease, including spinal cord injury, Parkinson's disease, and pancreatic beta cell insufficiency (diabetes mellitus). One promising therapy may be a clinical trial with human ES cells for congenital urea cycle disorder (https://www.amed.go.jp/en/news/release_20200521.html). A summary table as Table 2 is recommended.
(5) Fig 1A is just a method for the evaluation of mice cognitive functions. This figure may be omitted.
(6) Fig 1 B is important. However, the contribution of stem cells is unclear. The figure should be extensively revised to show specifically the targets and the mechanisms of stem cell therapy. For example, in the left-bottom figure (BACE and amyloid), how the stem cells affect this procedure should be shown. The molecular mechanism of how the stem cells decrease the beta-secretase activity should be illustrated.
(7) The problem of the neural circuit regenerated by neural stem cells should be discussed. The regeneration of the motor neurons may have less problems, because their main function is the contraction of muscles. The simple connection may be enough. However, in the brain, the complex neural circuits play a role in so-called higher brain function. The transplanted stem cells may only help to decrease the neural damages, but they may reconstruct the neural circuits. The treatment of mild or severe Alzheimer's disease may cause a completely different personality by regenerating new neural circuits. Although this point is briefly mentioned as " Another concern by Tan et al., is that the authenticity of the patient may be threatened after NSC implantation, as they may believe they have enhanced or changed the retention of particular memories [74]."(PDF page 20), a further discussion may be necessary.
(8) "N.A." in the tables may be "not available". It should be mentioned.
End of File
Round 2
Reviewer 1 Report
Hau Jun Chan et al. MS# ijms-1356132
Therapeutic potential of human stem cell implantation in Alzheimer’s disease
I would like to commend the authors for the nice job at revising their manuscript. Hau Jun Chan et al. have re-written extensive parts of the manuscript, modified the outline of the paper, and corrected several mistakes that created confusion in the original version the manuscript, in order to address the reviewers’ comments and criticisms. At this point, I support publication of the revised version of the authors’ review in the International Journal of Molecular Sciences.
Reviewer 2 Report
11 September 2021
Review on the manuscript titled “Therapeutic potential of human stem cell implantation and neurogenesis in Alzheimer’s disease” by Chan HJ et al., submitted to International Journal of Molecular Sciences (IJMS).
Manuscript ID: ijms-1356132
Dear Authors,
The authors present a narrative review article regarding stem cell therapy for the treatment of AD in preclinical and clinical studies, concluding that although there are limitations to overcome, the stem cell therapy is a potential treatment for AD.
Please reconsider the following parts:
- Pages 2, Abstract: Please clearly present that the manuscript is a narrative review.
- Page 2,3, Introduction: Please present the purpose of the manuscript in the end of Introduction.
- Pages 14-16, References: Please correct minor errors and missing parts.
The manuscript contains one figure, three tables, and 167 references. The authors revised the manuscript accordingly. The quality is substantially improved and the manuscript has become highly informative. The manuscript carries important value presenting the stem cell therapy of preclinical and clinical studies for the treatment of AD. I recommend this manuscript for publication after minor revision.
I declare no conflict of interest regarding this manuscript.
Best regards,
Masaru Tanaka, M.D., Ph.D.
